# Spray-Pyrolytic Tunable Structures of Mn Oxides-Based Composites for Electrocatalytic Activity Improvement in Oxygen Reduction

**Miroslava Varničić** [1]**, Miroslav M. Pavlović** [1,2,*]**, Sanja Eraković Pantović** [1]**, Marija Mihailović** [1]**, Marijana R. Pantović Pavlović** [1,2]**, Srećko Stopić** [3] **and Bernd Friedrich** [3]

[1] Department of Electrochemistry, Institute of Chemistry, Technology and Metallurgy, National Institute of the Republic of Serbia, University of Belgrade, Njegoševa 12, 11000 Belgrade, Serbia; mima.varnicic@gmail.com or varncic@ihtm.bg.ac.rs (M.V.); sanja@ihtm.bg.ac.rs (S.E.P.); marija.mihailovic@ihtm.bg.ac.rs (M.M.); m.pantovic@ihtm.bg.ac.rs (M.R.P.P.)

[2] Center of Excellence in Environmental Chemistry and Engineering-ICTM, University of Belgrade, Njegoševa 12, 11000 Belgrade, Serbia

[3] Process Metallurgy and Metal Recycling, RWTH Aachen University, Intzestraβe 3, 52072 Aachen, Germany; sstopic@ime-aachen.de (S.S.); bfriedrich@ime-aachen.de (B.F.)

\* Correspondence: mpavlovic@tmf.bg.ac.rs

**Abstract:** Hybrid nanomaterials based on manganese, cobalt, and lanthanum oxides of different morphology and phase compositions were prepared using a facile single-step ultrasonic spray pyrolysis (USP) process and tested as electrocatalysts for oxygen reduction reaction (ORR). The structural and morphological characterizations were completed by XRD and SEM-EDS. Electrochemical performance was characterized by cyclic voltammetry and linear sweep voltammetry in a rotating disk electrode assembly. All synthesized materials were found electrocatalytically active for ORR in alkaline media. Two different manganese oxide states were incorporated into a $Co_3O_4$ matrix, $\delta$-$MnO_2$ at 500 and 600 °C and manganese (II,III) oxide-$Mn_3O_4$ at 800 °C. The difference in crystalline structure revealed flower-like nanosheets for birnessite-$MnO_2$ and well-defined spherical nanoparticles for material based on $Mn_3O_4$. Electrochemical responses indicate that the ORR mechanism follows a preceding step of $MnO_2$ reduction to $MnOOH$. The calculated number of electrons exchanged for the hybrid materials demonstrate a four-electron oxygen reduction pathway and high electrocatalytic activity towards ORR. The comparison of molar catalytic activities points out the importance of the composition and that the synergy of Co and Mn is superior to $Co_3O_4/La_2O_3$ and pristine Mn oxide. The results reveal that synthesized hybrid materials are promising electrocatalysts for ORR.

**Keywords:** $MnO_2$; cobalt oxide $Co_3O_4$; perovskite materials; oxygen reduction in alkaline media; electrocatalyst; ultrasonic spray pyrolysis; Pt catalyst

## 1. Introduction

Economic development and extensive use of fossil fuels has led to fast depletion of energy resources. Hence, the development of clean energy storage and conversion devices, such as metal–air batteries, supercapacitors, fuel cells, and other renewable energy technologies, is in the main focus of numerous researchers and laboratories worldwide [1,2]. The efficiency of these energy devices mainly depends on the electrochemical oxygen reduction reaction (ORR) that occurs at the cathode side, as limiting reaction [3]. High activation barriers, poor rate capability, sluggish kinetics, and serious voltage gap of oxygen electrode reactions limit the performance of energy devices that rely on ORR [4–6].

Until now, the most investigated ORR catalysts have been based on noble metals such as Pt and Pt alloys, to achieve favorable reaction rates [7]. However, the high price and scarcity of the precious metals, inferior stability and sensitivity to CO poisoning, severely limit their widespread applications [2,3,8]. To overcome the above-mentioned

issues, lowering the amount of noble metals and exploring new catalytic materials for ORR have triggered extensive research interests.

Transition metals (TMOs) and organic macrocycles represent promising candidates as alternatives to noble metals as catalytic materials for ORR [9–14]. Among them, manganese oxides ($Mn_yO_x$) are of particular interest because of their prominent advantages of low cost, stability, environmental friendliness, abundance, and considerable catalytic activity toward oxygen reduction reaction [15–20]. Despite insufficient stability in acidic media, Mn-oxides can be applied as promising catalyst in air electrodes for both alkaline fuel cells and metal–air batteries. For example, various oxides have been studied, including perovskite-type, $\alpha$-$Mn_2O_3$ (bixbyite), $\beta$-$MnO_2$ (pyrolusite), $Mn_3O_4$ (spinel), $\alpha$-MnOOH (manganite), and simple Mn oxides [15–22], and the ORR was found to be highly dependent on the crystal structure of the oxides. Additionally, there are many studies showing manganese-oxide as highly promising material for the metal air batteries. For example, it has been reported the application of $MnO_2$ at the reduced graphene oxide as hybrid material in Mg-air battery [23], $MnO_2$ on graphene coated microfibers for Na–air battery [24], and Mn oxide framework for lithium–oxygen batteries [25]. It is possible to increase the activity of manganese oxide by tuning its crystal structure and morphology, doping, compositing, vacancy creation, and hydrogenation. The continuous improvement of the oxygen electrochemical activity of Mn-oxide is still ongoing work.

Cobalt oxide ($Co_3O_4$) is also one of the well-known materials that has been extensively studied as promising candidate and corrosion resistant ORR catalysts in alkaline media for fuel cells and metal–air batteries. Cobalt oxide belonging to the family of transition metal oxides, is able to display significant morphology modulated catalytic activity for ORR [26,27]. $Co_3O_4$ has spinal structure, with magnetic $Co^{2+}$ and non-magnetic $Co^{3+}$ at its tetrahedral and octahedral sites [27], that are significant for cobalt oxide catalytic activity. Porous $Co_3O_4$ nanoplates have been used as ORR catalyst for Zn–air batteries in alkaline medium [28], while flake-particles $Co_3O_4$ have been employed as catalysts for Li-$O_2$ batteries [29]. The structure of cobalt oxide has provided the abundant active sites together with ion and electron transport length, which eventually have improved the energy efficiency. Another investigation examines electrocatalyst for the oxygen reduction reaction based on a graphene-supported g-$C_3N_4$@cobalt oxide core–shell hybrid in alkaline solution with improved stability and activity, approaching to that of 20% Pt–C at the same potential [30]. It has been shown that the sole $CoO_x$ electrode exhibited only the two-electron mechanism with formation of hydrogen peroxide, rather than the four-electron mechanism, while the FCNTs electrode exhibited the two parallel mechanisms favoring four-electron mechanism only at higher overpotential. These results indicate the synergistic effect of the coupling between FCNTs and $CoO_x$ nanoparticles catalyzing the ORR via the direct four-electron mechanism. In another study, cobalt oxide nanocubes incorporated into reduced graphene oxide exhibited better electrocatalytic activity in terms of the current density, overpotential, and stability, compared to commercial Pt/C catalyst for the ORR in an alkaline medium [26]. Unfortunately, their ORR activity alone is generally poor. Thus, for further improvement, other metal atoms or carbon based materials have been incorporated into their catalysts structure [31–35].

La based perovskite materials are considered as a new class of materials in the mixed-oxide family and have attracted increasing attention for potential replacement of the noble metals. They have shown promising catalytic performance for ORR in alkaline media. The activity of these La-based oxides strongly correlated with the covalent bond strength between B-site cation and the oxygenated species. La is located in the middle of the octahedral structure and plays an important stabilizing role [36,37]. Additionally, $La_2O_3$ contain oxygen vacancies and interstitials with low oxygen vacancy energy, leading to low activation energy. Furthermore, the existing interlayer defect structure of the oxides is also helpful for the active oxygen adsorption, this all has a positive effect in catalyzing the ORR. However, even though lanthanum oxides have outstanding electronic structure, it is not electro-conductive, which limits its electrocatalytic capabilities and brings the necessity

to combine it with other oxides and carbon materials [38–40]. Due to its high potential as a stabilizing agent and high activity when mixed with other oxides, in this work it was utilized in the synthesis with Mn- and Co-oxide.

It was shown that combination of oxides, especially TMOs, and their structures, like spinel and perovskite-type oxides, could exhibit excellent ORR activities owing to the combination of metal elements, compared to single-metal oxides [7,41,42]. For example, Co-oxide nanoparticles modified with Mn-oxide nanotube have served as oxygen cathode catalyst for rechargeable zinc–air batteries [43]. La-, Co-, Mn-oxide prepared with carbon nanotubes (CNT) as composite has been successfully used as a bi-functional air electrode in Zn–air batteries [37]. The improved synergy effect has been reported in comparing to single oxide utilization. For example, various structures like honeycomb double-layer $MnO_2$/Cobalt doped for primary zinc–air batteries [44], 3D hollow sphere $Co_3O_4$/$MnO_2$-CNT [45], and core-shell $Co_3O_4$@$MnO_2$ [46] have been evaluated as bifunctional catalysts materials and applied for batteries and supercapacitors.

Therefore, we aimed to synthesize and investigate hybrid nanomaterials based on the Mn/Co/La oxides of ordered structure generated by ultrasonic spray pyrolysis (USP) as electrocatalyst for ORR. USP technique was chosen for the synthesis of these materials as it allows a simple single step approach of synthesizing nanomaterials with precisely controllable morphologies and chemical compositions. Different compositions and morphologies were synthesized depending on USP temperature and tested.

One of the issues to be considered is that the future ORR materials should not contain rather electrochemically unstable carbonaceous materials as support. The investigated $MnO_x$-$Co_3O_4$ TMO hybrid electrode nanomaterials were carbon free. The influence of manganese oxide type on the ORR as well as difference in composition of Mn and Co was evaluated. The second important issue that we tackled is that the materials should be synthesized using a low-cost, simple technique that can be easily applied also for the large scales such as USP.

## 2. Experimental

### 2.1. Chemicals

Lanthanum(III)nitrate hexahydrate La(NO$_3$)$_3$ $\times$ 6H$_2$O (99.9% rare earth oxide), manganese(II)nitrate tetrahydrate Mn(NO$_3$)$_2$ $\times$ 4H$_2$O (99%), and cobalt(II)chloride hexahydrate CoCl$_2$ $\times$ 6H$_2$O (99%) were used during the synthesis process and were purchased all from Alfa Aeser, US. For the comparison, commercial manganese (IV) oxide, $MnO_2$, was used and obtained from Sigma-Aldrich (Saint Louis, MO, USA). Potassium hydroxide and Nafion 117 solution (5 wt.%) were purchased from Sigma-Aldrich. All chemicals were of analytical reagent grade and all solutions were prepared using ultrapure water from Millipore.

### 2.2. Material Synthesis and Electrode Preparation

2.2.1. Material Synthesis

The synthesis of Co/Mn/La oxide hybrid materials was performed by single-step ultrasonic spray pyrolysis process. The solution for the material synthesis was prepared by mixing starting precursor solutions to give desired stoichiometric mole ratios La:Co:Mn = 3:5:10. The ratio of La:Co:Mn of 3:5:10 was chosen based on previous research [47,48], since we wanted to investigate the influence of Mn in mixture for ORR reaction. Aqueous 0.1 M solutions of La(NO$_3$)$_3$, Mn(NO$_3$)$_2$ and CoCl$_2$ were used as precursors. The USP conversion temperature was adjusted and controlled using a thermostated furnace. All powders were synthesized by ultrasonic spray pyrolysis in the equipment with horizontal nebula flow.

Nebula generation from the prepared solutions of precursors took place in an ultrasonic atomizer (Gapusol 9001, RBI/France) with an ultrasonic nebulizer (Prizma Kragujevac, Serbia) to create an nebula-born aerosol [47,48]. The nebula with droplets having a diameter of around 2.3 μm was produced with an ultrasound frequency of 2.5 MHz. The nebulization/aerosol generation was carried out in O$_2$/N$_2$ atmosphere as carrier gas,

having $O_2$ to $N_2$ in volume ratio of 2:1 and continuous flow rate of 3 $dm^3$ $min^{-1}$. The synthesis temperatures were set to 500, 600, or 800 °C.

### 2.2.2. Electrode Preparation

For the electrochemical measurements, the catalyst-modified surface of electrodes was prepared from the glassy carbon disc (Pine Research Instrumentation USA, 5 mm). Prior to the use, glassy carbon disc was polished with alumina slurry kit (Pine Research Instrumentation, Durham, NC, USA) of different grades, and then cleaned ultrasonically in ethanol and water.

The electrodes were prepared in the following way. Firstly, 5 mg $mL^{-1}$ of water suspension of the USP-synthesized powder was agitated in an ultrasonic bath for 30 min in order to form homogeneous ink. Then, 20 µL of the ink were cast by micropipette onto the glassy carbon disc and left to air-dry for 2 h. In the next step, 10 µL of Nafion solution (100:1 diluted commercial Nafion solution) were pipetted onto the catalyst-covered GC disc, as binding agent, and left to dry at room temperature.

### 2.3. Measurements

### 2.3.1. Material Characterization

Structural and phase analysis of the synthesized materials was investigated by X-ray diffraction (XRD). The measurements were undertaken on a Philips PW 1050 powder diffractometer with Ni-filtered CuKα radiation at room temperature and scintillation detector within the range 10–82° in steps of 0.05° with the scanning rate of 5 s/step.

Scanning electron microscopy (SEM) with an energy dispersive X-ray spectroscopy (EDS) were employed to analyze morphology and element composition of porous Mn/Co/La oxide hybrid materials. Scanning electron microscope (Zeiss DSM 982 Gemini; Vega TS 5139MM Tescan, Brno, Czech Republic) was employed for the examination of obtained particles on a different magnification level providing different information on the morphology and particle shape and size. The elemental composition was determined by EDS with Si(Bi) X-ray detector connected to SEM and a multi-channel analyzer.

### 2.3.2. Electrochemical Measurements

Electrochemical measurements were performed using BioLogic potentiostat (BioLogic SAS, SP-240, Grenoble, France). The electrocatalytic properties of the hybrid materials were checked by means of cyclic voltammetry (CV) and linear sweep voltammetry (LSV), using the scan rate of 50 and 2 mV $s^{-1}$, respectively. In order to check the material activity for oxygen reduction reaction, 3-electrode set-up, using a rotating disk working electrode was employed. The GC modified with synthesized materials as described in Section 2.2.2 were used as working electrode, while saturated calomel (SCE) and Pt electrode were employed as the reference and counter electrode, respectively. All the potentials presented are referred to the SCE. The supporting electrolyte was a 0.1 M aqueous KOH solution. All electrochemical experiments were performed at 25 °C under nitrogen or oxygen atmosphere at 600, 800, 1000, 1500, or 2500 rpm rotation of the working electrode. Prior to every experiment, either in $N_2$ or $O_2$ atmosphere, the gas was bubbled through the electrolyte for at least 20 min.

## 3. Results and Discussion

### 3.1. XRD Analysis

The hybrid nanomaterials based on rare earth/transition metal oxides were synthesized with facile and cost-effective USP procedure, bearing in mind the methodology as follows. One group of electrocatalyst is based on manganese, cobalt, and lanthanum metals and the effects of three different USP temperatures were investigated (500, 600, and 800 °C). The other group of materials was prepared by the same synthesis procedure, but without manganese component—it was only based on cobalt and lanthanum oxide, in order to investigate the influence of USP-synthesized Mn oxide within hybrid oxide electrocatalysts.

For the sake of comparison, the electrocatalytical performance of commercial manganese oxide was also included in the investigations.

The crystalline structures of the hybrid materials were revealed by XRD analysis. X-ray diffraction patterns of synthesized materials are presented in Figure 1. As can be seen in Figure 1a, the materials synthesized at 500 and 600 °C are composed mainly of manganese (IV) oxide in the form of birnessite, also denoted as $\delta$-$MnO_2$, as defined by main diffraction peaks at $2\theta$ of 12.4, 25.3, 37, and 66°. The specific XRD peaks correspond to a card no.: JCPDS 00-043-1456 ($MnO_2$). Birnessite is reported as 2D layered manganese oxide with lamellar structure consisting of edge-sharing $MnO_6$ octahedra, and is considered to be the most active phase for ORR among other crystalline structures of $MnO_2$ [49–51]. In addition, the diffraction peaks at 18.9, 31.2, 45, and 59° reveal the presence of $Co_3O_4$ (JCPDS card no. 01-080-1535). However, La oxide or other compounds which should indicate the La presence are not evidenced. It follows that La is present as poorly crystalline or amorphous lanthanum compound(s). These "La-hided" states of Co-La oxide hybrids corresponds to our recent findings [48].

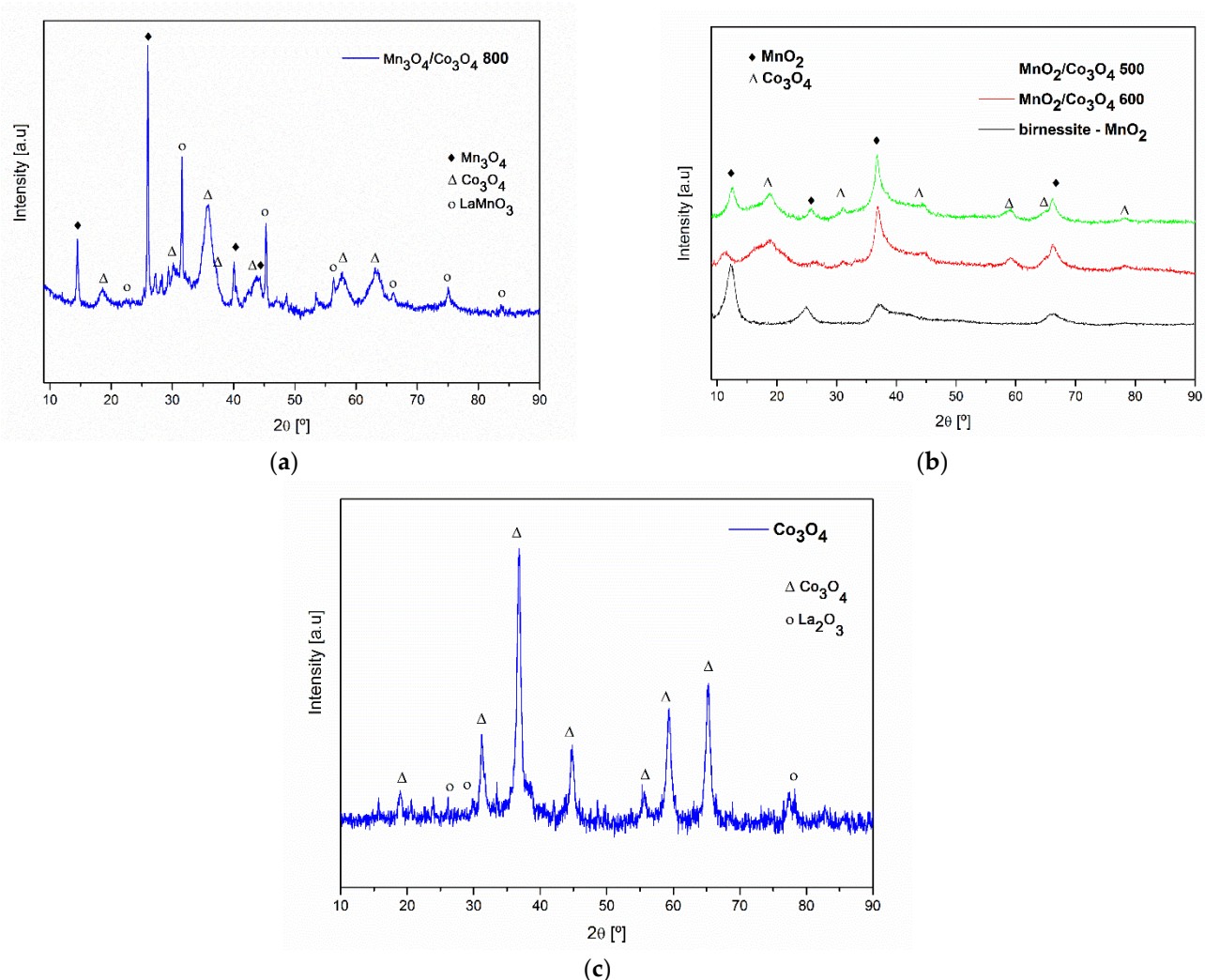

**Figure 1.** XRD patterns of (**a**) Mn/Co/La oxide hybrid materials synthesized on 500 and 600 °C, and commercial $MnO_2$; (**b**) Mn/Co/La oxide material synthesized at 800 °C; (**c**) Co/La oxide hybrid material.

XRD pattern of the material sample synthesized at 800 °C is presented in Figure 1b. The presence of $Co_3O_4$ is clearly confirmed; however, the other type of Mn oxide-$Mn_3O_4$ is formed. It is evidenced by well-resolved peaks at the positions 14.4, 26.2, 40.1, and 44°

(JCPDS card no. 03-06502776). Additionally, the diffraction peaks related to the JCPDS card no. 01-075-0440 confirms the formation of the perovskite structure-LaMnO$_3$. It can be seen that XRD peaks in Mn-containing compounds synthesized at 800 °C are sharper than those at lower temperatures, which indicate highly crystalline nature of those compounds. The formation of different manganese oxide types at different temperatures is expected. It has been already reported that under these conditions Mn changes its form from Mn(IV) oxide to Mn(II, III) oxide [52]. Synthesized hybrid materials are denoted as MnO$_2$/Co$_3$O$_4$-500, MnO$_2$/Co$_3$O$_4$-600, and Mn$_3$O$_4$/Co$_3$O$_4$-800 in the further text, according to XRD findings.

The structural and phase characteristics of the Mn-free synthesized material, based on cobalt and lanthanum compounds, is presented in Figure 1c. XRD peaks clearly confirm formation of Co$_3$O$_4$. In addition, weak peaks at positions 26.2°, 29.8°, and 78° indicate the formation of La$_2$O$_3$, although it can be assumed that this oxide is present mainly in an amorphous form. Thus, this catalytic material is denoted as Co$_3$O$_4$/La$_2$O$_3$ in further text.

### 3.2. SEM and EDS Characterization

The morphology of as-synthesized catalytic materials was investigated by SEM. Depending on the preparation temperature the samples morphology appears different, as shown in Figure 2. For the materials prepared at the temperatures 500 and 600 °C, the results indicate formation of spherical grains, Figure 2a,c, with a petal-like structure discovered at the higher-resolution images (Figure 2b,d). It can be seen that the petal-structured grains, having the size of around 2 µm, are built from numerous nanosheets. The nanosheets appear finer and more densely packed at higher USP temperature (Figure 2b,d). The very similar structures were observed by Che et al. reporting the core-shell microspheres composed of Co$_3$O$_4$@MnO$_2$ with flower-like structured Co$_3$O$_4$ as the core onto which MnO$_2$ nanosheets have been subsequently grown [46]. This typical flower-like morphology of birnessite-MnO$_2$ forming micro/nanospheres has been also reported to have high surface area that might exhibit fast electrode kinetics and good stability. MnO$_2$ nanosheets are thus recognized as excellent candidates for electrocatalytic materials for electrochemical oxygen reactions [50,51,53].

On the other hand, the Mn$_3$O$_4$/Co$_3$O$_4$ material synthesized at 800 °C has homogenous dense structure as presented in Figure 3. The enlarged SEM image reveals well-defined submicron particles of the catalytic material (Figure 3b). Similar change in morphology, leading to the formation of defined particles instead of flower-like structure, has been reported and assigned to the presence of Mn$_3$O$_4$ type of Mn oxide, in comparison to the distinguished nanosheets typical for MnO$_2$ [54,55]. The SEM images of Co$_3$O$_4$/La$_2$O$_3$ catalytic material shows highly agglomerated particles with irregular shapes and various sizes ranging from nano- to several µm (Figure 3d). The porous agglomerated particles of Co$_3$O$_4$ material have also been reported, providing large micro- and mesoporous surface [56].

The elemental composition, done by energy dispersive X-ray spectroscopy (EDS), confirms the presence of Mn, Co, La, and O with the atomic ratios presented in Table 1. It can be seen that for MnO$_2$/Co$_3$O$_4$-500, MnO$_2$/Co$_3$O$_4$-600, and Mn$_3$O$_4$/Co$_3$O$_4$-800 materials, the obtained atomic ratio of Mn:Co is close to 2:1, which is in accordance to the projected atomic ratio used for synthesis. On the other hand, the atomic ratio between lanthanum, cobalt, and manganese is smaller than projected ratio. This anticipates that the material is likely structured as separated phases of the oxides of well-resolved crystalline state, MnO$_2$ and Co$_3$O$_4$, covering poorly crystalized La compounds, as found by XRD, which can mask its EDS response. Finally, it was seen in XRD patterns that Co$_3$O$_4$/La$_2$O$_3$ catalytic material exhibits diffraction peaks similar to La$_2$O$_3$ card, which can cause the apparently hidden XRD state of La$_2$O$_3$ by crystalline Co$_3$O$_4$.

### 3.3. Electrochemical Characterization

To study the electrocatalytic performances of synthesized Mn/Co/La-hybrid materials for oxygen reduction reaction, three-electrode half-cell design was used. Electrodes were

prepared with catalytic material as described above in Section 2.2.2 and used as working electrodes in an RDE system to provide constant hydrodynamic conditions.

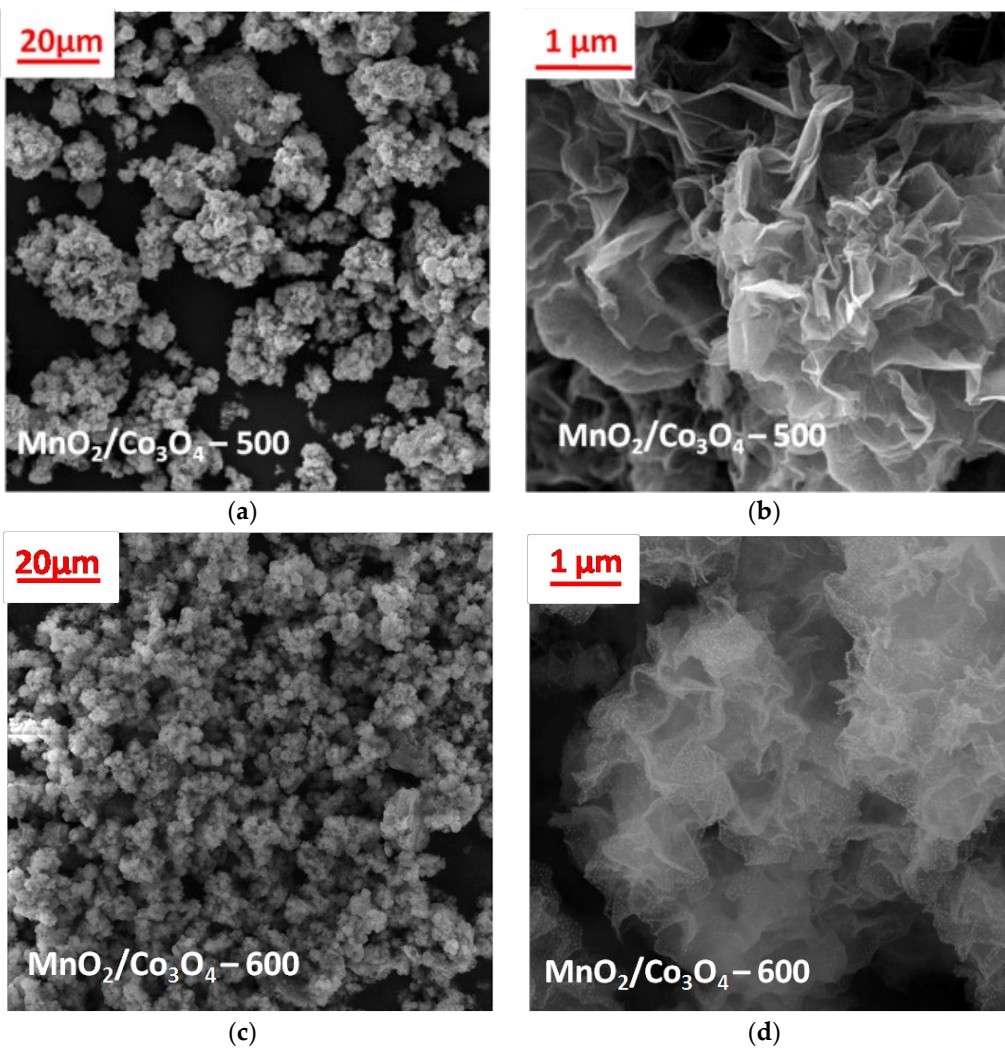

**Figure 2.** SEM images of (**a**,**b**) MnO$_2$/Co$_3$O$_4$-500, (**c**,**d**) MnO$_2$/Co$_3$O$_4$-600; at lower and higher magnifications.

Cyclic voltammograms were recorded in the potential window between −0.9 and 0.6 V, at the scan rate of 20 mV s$^{-1}$. Figure 4 presents CV responses of synthesized materials and pristine MnO$_2$ electrode in deaerated 0.1 M KOH. The cyclic voltammograms of MnO$_2$/Co$_3$O$_4$ electrodes exhibit almost featureless shape with capacitive current showing some reversible charge transfer processes at the potentials positive to 0.1 V, with counterparts negative to −0.1 V. This behavior is in accordance with the literature results for other MnO$_2$ nanostructures electrodes reporting the similar behavior in nitrogen atmosphere as well as for pristine MnO$_2$ [44,51,57]. The redox processes appeared suppressed upon increase in synthesis temperature. On the other hand, CV behavior of Co$_3$O$_4$/La$_2$O$_3$ hybrid material shows fully reversible redox transitions of much higher currents at the potentials positive to −0.1 V, which can be assigned to redox transitions of Co. This CV performance has been reported as typical behavior of cobalt oxide nanoparticles [58]. The shape of CV curves of Co$_3$O$_4$/La$_2$O$_3$ is different in comparison to that of MnO$_2$/Co$_3$O$_4$ materials, with considerably higher capacitive currents. It seems that CV fingerprints follow the registered structural organization of the oxides in the Co$_3$O$_4$/La$_2$O$_3$ material. Cobalt oxide particles dictate the CV behavior of Co$_3$O$_4$/La$_2$O$_3$ material in a way to resemble completely the redox processes of pure Co$_3$O$_4$. On the other hand, Mn-based materials are of CV behavior similar to pristine MnO$_2$, since the particle surface composition is of twice as much

as nominal loading of manganese with respect to cobalt. CV response of $MnO_2/Co_3O_4$ synthesized at lower temperature ($MnO_2/Co_3O_4$-500) strives for the shape more similar to that of $Co_3O_4$. This could be related to the more spaced petals (Figure 2b) with respect to dense appearance of petals at higher synthesis temperature ($MnO_2/Co_3O_4$-600, Figure 2d). It follows that $C_{O3}O_4$ contributes more to CV response through the more spaced $MnO_2$-rich petals (Table 1). Additionally, it seems that the absence of petal-like structure and transition from $MnO_2$ to $Mn_2O_3$ (Figures 1b and 3d) in the case of $Mn_2O_3/Co_3O_4$-800 does not affect much the CV response of rather low-current featureless characteristics.

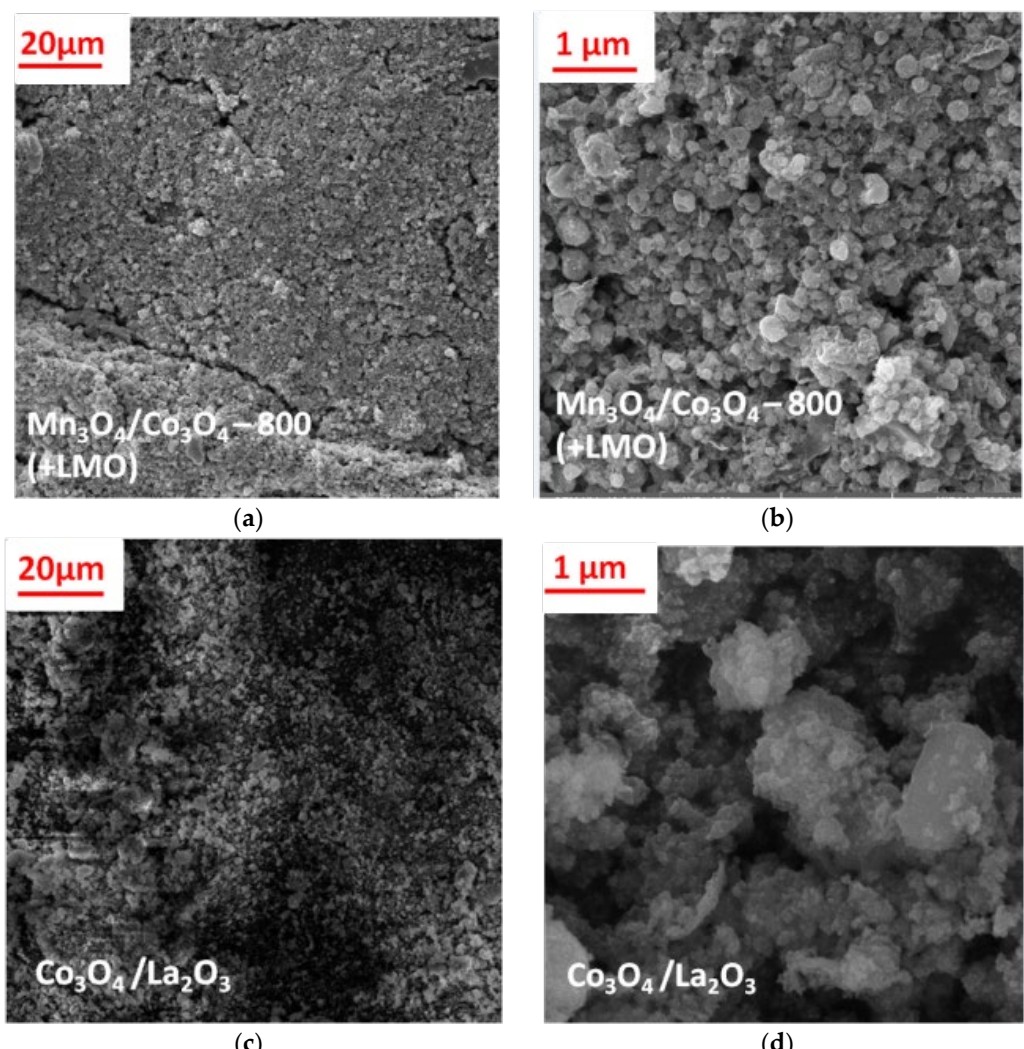

**Figure 3.** SEM images of (**a,b**) $Mn_3O_4/Co_3O_4$-800 and (**c,d**) $Co_3O_4/La_2O_3$; at lower and higher magnifications.

**Table 1.** Element analysis of synthesized materials for EDS analysis (at. %).

| Element | Sample | | | | |
|---------|--------|--------|--------|--------|--------|
| | $MnO_2/Co_3O_4$-500 | $MnO_2/Co_3O_4$-600 | $Mn_3O_4/Co_3O_4/LaMnO_3$-800 | $Co_3O_4$ (+$La_2O_3$) | $MnO_2$ |
| O | 68.71 | 62.92 | 61.78 | 69.77 | 60.29 |
| La | 2.04 | 3.78 | 3.11 | 2.26 | / |
| Co | 9.88 | 11.24 | 12.98 | 27.96 | / |
| Mn | 19.37 | 22.06 | 22.13 | / | 32.43 |

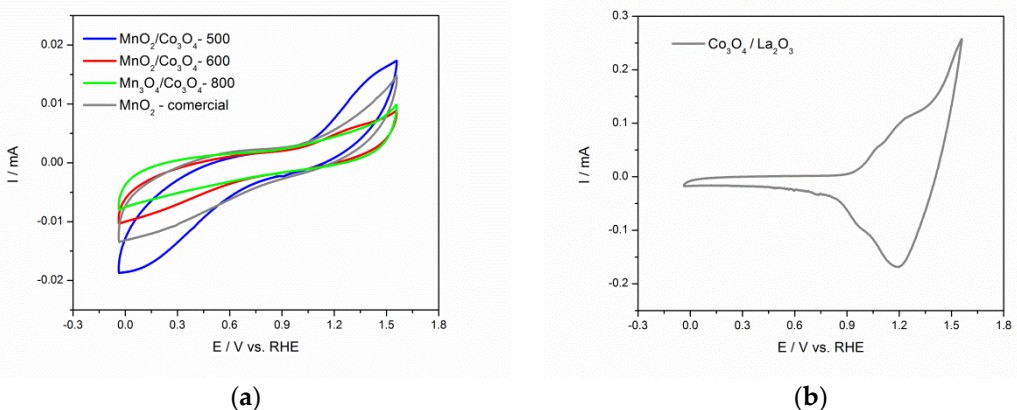

**Figure 4.** CV performance of synthesized catalytic materials (**a**) comparison between Mn-based materials and commercial $MnO_2$, (**b**) Co-based electrode; 0.1 M KOH, $N_2$ atmosphere, 20 mv s$^{-1}$.

The electrocatalytic activities of the prepared nanocatalysts for ORR were evaluated by means of LSV in $O_2$-saturated alkaline electrolyte. Figure 5 presents ORR electrochemical performances of Mn/Co/La oxides synthesized at three different temperatures. As can be seen, when the electrolyte was saturated with $O_2$, remarkable reduction currents are observed, which introduces the synthesized materials as ORR-active. The onset potential of all electrodes was approx. $-0.3$ V vs. SCE, which is competitive to other TMO based materials [56]. The electrode material at 500° and 600° show similar curve shape and activity, which is expected for similar flower-like birnessite $MnO_2$, having similar electrochemical activities between 1 and 2.5 mA cm$^{-2}$ (Figure 5a) [51]. The $Mn_3O_4/Co_3O_4$ material, obtained at 800 °C, exhibits higher ORR activity than $MnO_2/Co_3O_4$-500 and $MnO_2/Co_3O_4$-600 materials. This can be ascribed to the presence of $Mn_3O_4$ as catalyst with mixed oxidation state (+2, +3) in comparison to the $MnO_2$ with +4 oxidation state of Mn [20]. Additionally, as it is registered by SEM, two different manganese types provide different morphologies. Consequently, the ORR current increases continuously for $Mn_3O_4/Co_3O_4$-800, whereas the reduction on $MnO_2/Co_3O_4$-500 and -600 appears stepped, with a transition around $-0.7$ V. This could be the indication of different ORR mechanisms on $MnO_2/Co_3O_4$ and $Mn_3O_4/Co_3O_4$.

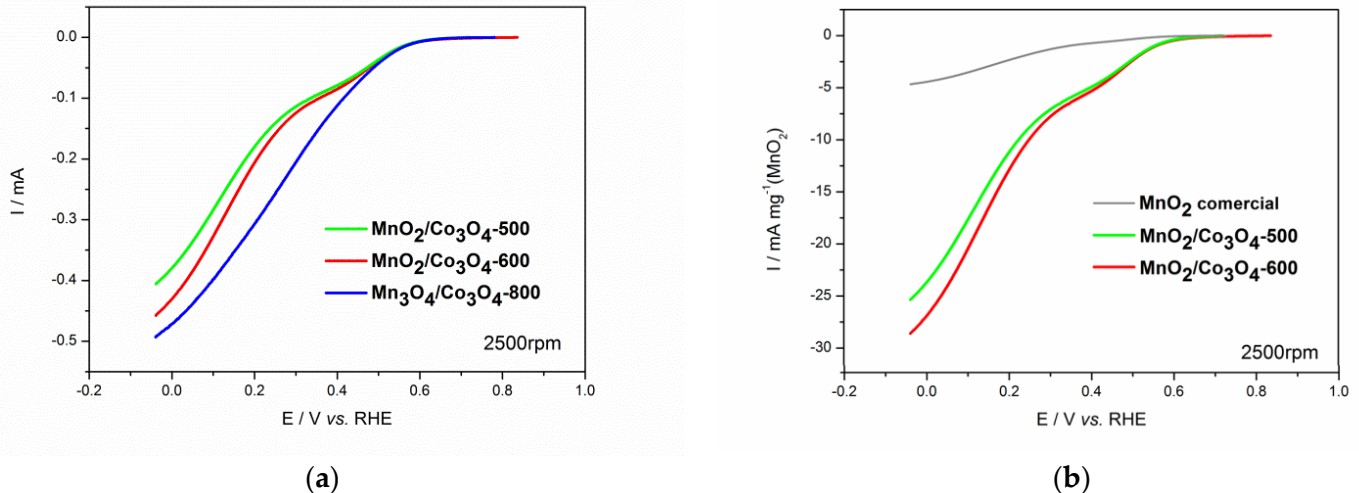

**Figure 5.** Comparison of ORR activities of Mn-based hybrid materials: (**a**) at different synthesis temperature and (**b**) commercial $MnO_2$ for ORR per mass of $MnO_2$.

For the sake of comparison to pristine $MnO_2$, electrode activities calculated per mass of $MnO_2$ (mass activity) was performed, as presented in Figure 5b. It can be seen that nanostructured $MnO_2/Co_3O_4$ materials are of significantly improved ORR activity compared to

the pristine manganese oxide. This proves the validity of the hybrid oxides approach of ordered structure for the synthesis of materials for ORR.

In order to study ORR activity further, the series of polarization curves (LSV) in saturated oxygen atmosphere were recorded at different electrode rotating rates between 600 and 2500 rpm. As can be seen in Figure 6a, ORR current is increasing with the increase of the rotation rate at higher overpotentials, due to the improved mass transfer. However, the first reduction step (positive to −0.7 V) for $MnO_2/Co_3O_4$-500 and -600 appears negligibly dependent on rotation rate. It follows that corresponding process(es) are not directly related to ORR, but to partial reduction of the material induced by the presence of oxygen (please see Equations (3)–(7)).

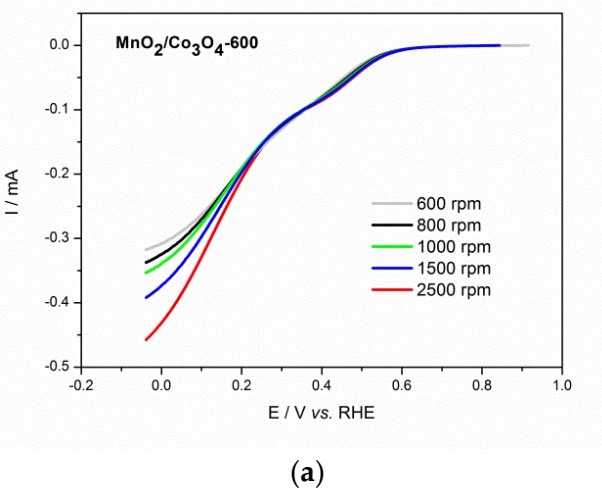

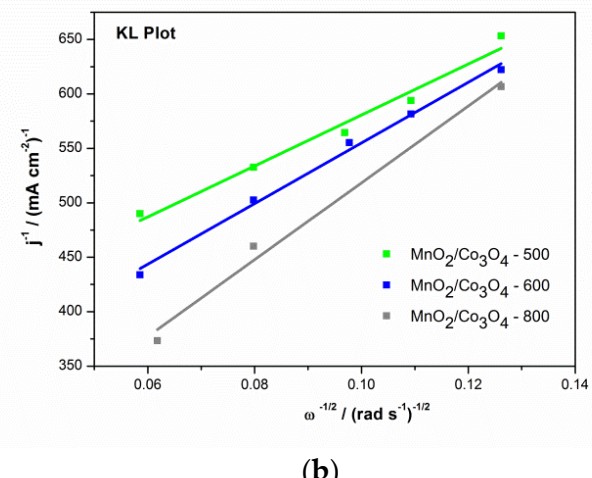

(a)  (b)

**Figure 6.** LSV performance of synthesized materials: $MnO_2/Co_3O_4$-600 (**a**) at different electrode rotation rates; (**b**) KL-plot of the synthesized materials; 0.1 M KOH, $O_2$ atmosphere.

The number of the electrons that are involved in oxygen reduction reaction is an important parameter for evaluating the catalytic performance of the synthesized materials. Therefore, the ORR was further analyzed using Koutecky–Levitch (KL) equation (Equation (1)). The corresponding linear fit that is presented in Figure 6b at E = −1 V, was used to calculate the number of electrons transferred during the oxygen reduction with synthesized materials. In the equation:

$$\frac{1}{j} = \frac{1}{j_L} + \frac{1}{j_k} = \frac{1}{B\omega^{0.5}} + \frac{1}{j_k} \tag{1}$$

$$B = 0.62nFC_0D^{2/3}v^{-1/6} \tag{2}$$

where $j$ corresponds to the measured current density, $n$ is the overall number of electrons exchanged, $F$ stands for Faraday constant ($F$ = 96486 C $mol^{-1}$), $C_0$ is the oxygen concentration in 0.1 M KOH (typically $C_0$ = 1.2 × $10^{-6}$ mol $cm^{-3}$), $D$ is oxygen diffusion coefficient (typically $D$ = 1.9 × $10^{-5}$ $cm^2$ $s^{-1}$), $v$ is the kinematic viscosity of the solution ($v$ = 0.01 $cm^2$ $s^{-1}$); surface area of the electrode used to calculate current density is $A$ = 0.196 $cm^{-2}$.

It has been reported that the ORR in alkaline media can proceed either via two-electron pathway, which involves the formation of hydrogen peroxide as an intermediate, or via direct four-electron reduction pathway where oxygen is directly reduced to $OH^-$ [59]. Generally, direct four-electron transfer pathways are more desirable than the partial reduction pathway since it provides a higher rate for ORR. Although the ORR mechanism on Mn-oxide is still not fully understood, the possible pathway suggests the reactions described:

$$MnO_2(s) + H_2O + e^- \leftrightarrow MnOOH(s) + OH^- \tag{3}$$

$$2MnOOH(s) + O_2 \leftrightarrow 2(MnOOH\ldots.O)\ (s) \tag{4}$$

$$MnOOH(s) + O_2 \leftrightarrow MnOOH \ldots .O_2,_{ads} (s) \tag{5}$$

$$(MnOOH \ldots .O) + e^- \leftrightarrow MnO_2(s) + OH^- \tag{6}$$

$$MnOOH \ldots .O_2,_{ads} (s) + e^- \leftrightarrow MnO_2(s) + HO_2^- \tag{7}$$

The overall reaction of Equations (3), (4), and (6) equals to the four-electron reduction process, whereas the summary of Equations (3), (5), and (7) results in an overall two-electron transfer mechanism oxygen reduction. In the first step, $Mn^{4+}$ is reduced to $Mn^{3+}$, which is followed by adsorption and reduction of oxygen. Hence, the promotion of $Mn^{3+}$ generation could lead to more effective ORR over stronger oxygen adsorption and accelerated $O_2$ reduction to $OH^-$, which results in overall increase in catalytic activity [49]. Furthermore, it has been reported that porous structure can also stabilize $Mn^{3+}$ species at the particle surface [36]. For the synthesized materials, Figure 6b reveals that the number of transferred electrodes, calculated from the slope, for $MnO_2/Co_3O_4$-500 and -600 catalytic materials are 3.6 and 3.89 which is close to 4 indicating that ORR catalyzed by those materials proceeds via quasi-four-electron ORR mechanism. However, the overall electron transfer number of $Mn_3O_4/Co_3O_4$-800 material is calculated to be 3, indicating that both of the suggested schemes coexist in the catalyzing process [60]. Even though the higher oxidation states of Mn ($Mn^{4+}$ and $Mn^{3+}$) are considered as crucial for manifestation of Mn cation defects and oxygen vacancies that are important as catalytically active sites, the morphology of samples at 500, 600, and 800, as well as their mutual interaction with Co-oxide, also plays an important role in catalytic activity.

The $Co_3O_4/La_2O_3$ material was also checked for the ORR performance at various rotation speeds, as presented in Figure 7. The ORR on this hybrid material starts at approx. $-0.35$ V vs. SCE, which is in accordance to other $Co_3O_4$ reported nanomaterials, but still is considerably negative if compared to the commercial 20 wt.% Pt@XC-72 catalyst ($-0.16$ V) [56]. On the other hand, the current of approx. $-3.3$ mA cm$^{-2}$ is higher in comparison to the performance of similar nanomaterials, found as $-2.5$ [56], $-1.5$ [43], and $-1$ mA cm$^{-2}$ [58] at similar electrode potentials. ORR on synthesized $Co_3O_4/La_2O_3$ was studied also using a KL plot shown in Figure 7b. The number of electrons transferred, calculated based on KL equation, is 3.7 suggesting predominantly the pathway of direct four-electron reduction of oxygen. This is fairly comparable to the state-of-the-art electrode based on Pt (20% wt. Pt@XC-72), which is reported to be between 3.8 and 4.03 [56].

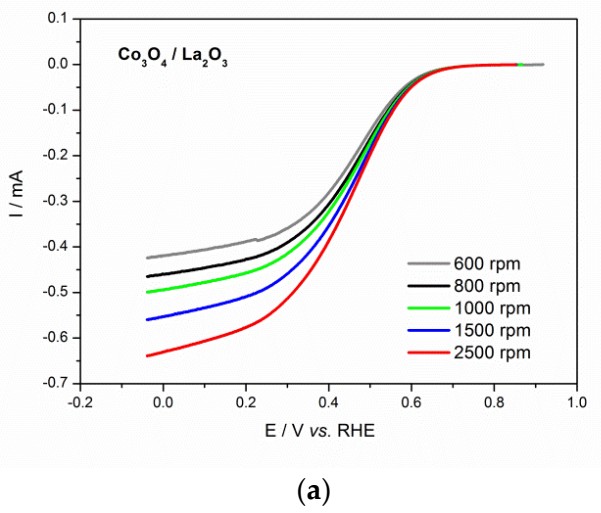

**(a)**

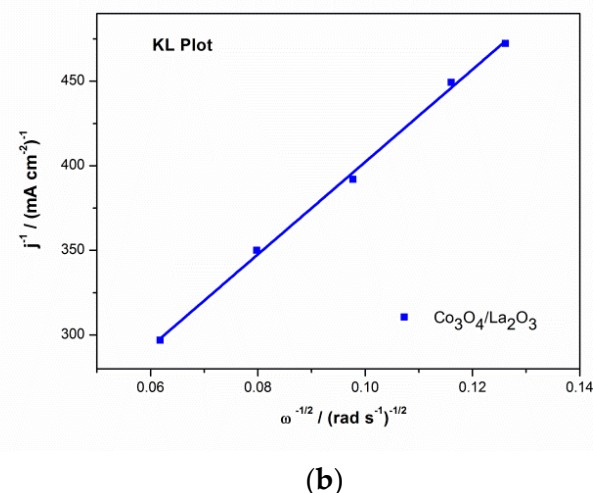

**(b)**

**Figure 7.** Reduction of $O_2$ at the Co-based electrode (**a**,**b**) KL plot for $Co_3O_4/La_2O_3$.

Finally, the comparison of all synthesized oxide combinations as catalytic materials is presented in Figure 8. As can be seen, $Co_3O_4/La_2O_3$ catalyst outperforms $MnO_2/Co_3O_4$ and $Mn_3O_4/Co_3O_4$ materials, in comparison to the onset electrode potential, as well as in electrocatalytic activity toward ORR in the studied potential region. Similar behavior has

been reported by Xu et al. showing that $Co_3O_4/La_2O_3$ supported by carbon nanotubes (CNT) has shown better activity in comparison to the $MnO_2/Co_3O_4$–CNT. Although CNT have been used to increase catalytically active surface area, the activities of the studied hybrid materials are in the range 1.8–4 mA cm$^{-2}$ that is comparable to our hybrid materials but without addition of a carbon support [37].

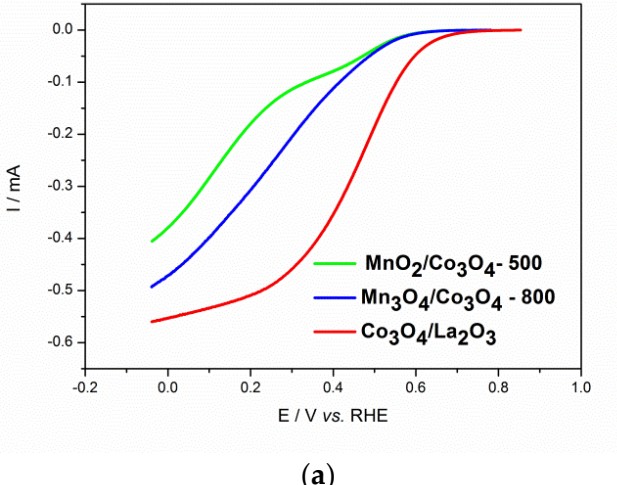 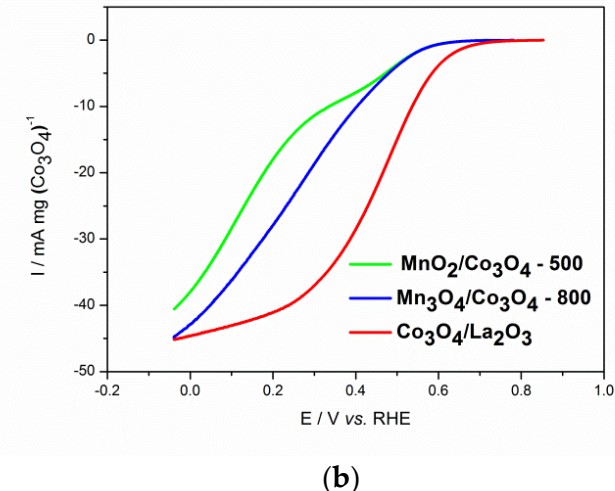

(**a**)                                                        (**b**)

**Figure 8.** Comparison of (**a**) synthesized materials in mA (**b**) calculated per mass of Co.

As mentioned above, the mechanism on the synthesized electrode most probably proceeds with an additional MnOOH reaction step. It has been reported that this reaction is unfavorable due to the strength of the Mn-O bond which makes the initial reduction more difficult. This leads to lower electrocatalytic activity of the electrodes with high percentage of Mn oxide of birnessite type. It is also speculated that the crystal phase (channel structure) can be another determining criterion for ORR kinetics at manganese oxides.

Additionally, the observed larger inner-spacing between nanoparticles in $Co_3O_4/La_2O_3$ (Figure 3) in comparison to dense structure of $MnO_6$ octahedral sheet of birnessite $MnO_2$, provides more active surface area that is more accessible to the electrolyte (reactant). In addition, it is reported that $Co_3O_4$ has high affinity toward $O_2$ molecules, which enables better oxygen transport within its porous structure [61].

This all contributed to the observed behavior that the $Co_3O_4/La_2O_3$ electrode exhibits better catalytic activity in comparison to the $MnO_2/Co_3O_4$ electrodes. Additionally, Du et al. stated that $Co_3O_4$ nanoparticles-modified $MnO_2$ electrodes have much lower ORR activity in comparison to pure $MnO_2$ nanomaterial due to partial occupation of active sites on $MnO_2$ by $Co_3O_4$ [43]. This likely can be another reason for the lower catalytic activity of our $MnO_2/Co_3O_4$ oxides catalysts. Although the synergic effect of $MnO_2$ and $Co_3O_4$ oxides has been reported in many publications to increase ORR activity [43,45], the investigation of the parameters such as the composition, crystalline structure, and morphology are to be investigated in order to propose the most probable synergy mechanism.

Since the synthesized hybrid materials have different compositions, and consequently structures, the kinetic comparison would be more informative if would be presented as activity (currents) per mol of $Co_3O_4$ and $MnO_2$. Figure 9 presents the comparison of molar activities with respect to $Co_3O_4$ for the samples containing Co oxide, and with respect to $MnO_2$ for the samples synthesized at 500 and 600 °C, taking into account the compositions found by EDS (Table 1). As can be seen from Figure 9a, the $Co_3O_4/La_2O_3$ electrode is of higher activity at the beginning (at lower overpotentials) due to a different ORR reaction mechanism occurring in comparison to the Mn-based electrodes (additionally involves the Mn oxide electrochemical transformations). However, in the region of higher overpotentials that are more relevant for the electrochemical devices (fuel cells, batteries), the electrodes based on synergy of Mn/Co oxides outperform the electrode without Mn ($Co_3O_4/La_2O_3$). This result is somewhat contrary to the presented performance of the current calculated

per mass (Figure 8), which indicated the electrode without Mn ($Co_3O_4/La_2O_3$) as the one of best performances. In addition, Figure 9b shows that trends of molar activities of $MnO_2$-containing electrodes are similar to those of mass activity (Figure 5b). This clearly emphasizes the importance of the calculations to take into consideration the mole fractions in hybrid materials of different compositions in order to quantify the synergy effects.

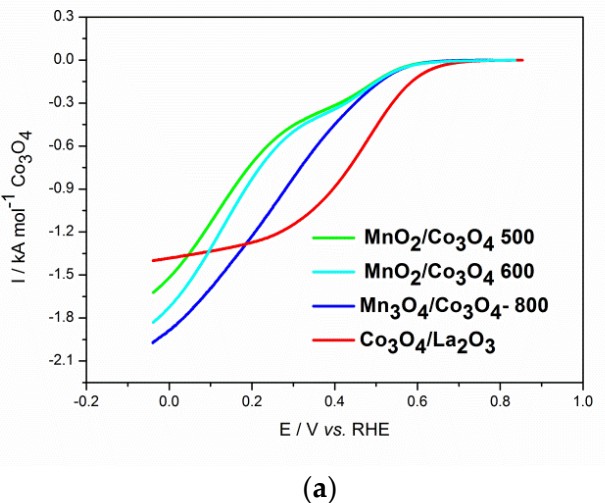

(**a**)

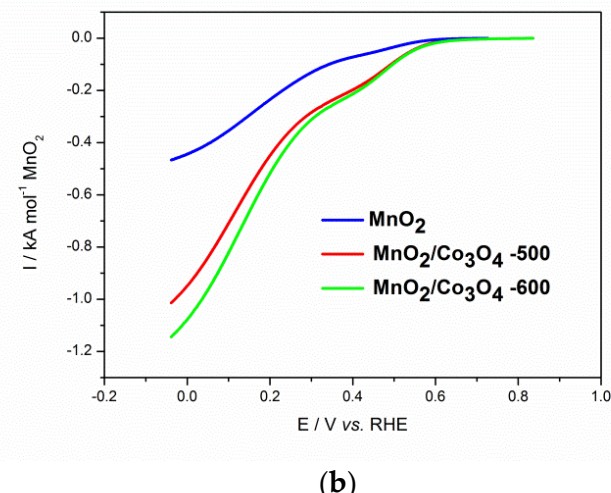

(**b**)

**Figure 9.** Comparison of all materials (**a**) per mol of Co oxide and (**b**) per mol of Mn oxide.

### 4. Conclusions

In summary, highly active electrocatalysts based on Mn and rare earth oxides with $Co_3O_4$ for ORR, have been successfully synthesized using the ultrasonic spray pyrolysis process. For the sake of comparison, the hybrid materials with and without Mn oxides were investigated.

It was shown that different Mn oxides were well incorporated in the $Co_3O_4$ matrix. Different synthesis temperatures led to the formation of two different manganese oxides-birnessite type δ-$MnO_2$ at 500 and 600 °C, and manganese (II, III) oxide-$Mn_3O_4$ at 800 °C. The catalysts morphology has been also affected by the state of Mn oxide. SEM images reveal flower-like nanosheets for hybrid materials with birnessite-$MnO_2$ and well-defined spherical nanoparticles for material based on $Mn_3O_4$ and for the material based only on Co as catalysts ($Co_3O_4/La_2O_3$). The electrochemical performance of $MnO_x/Co_3O_4$ and $Co_3O_4/La_2O_3$ demonstrate a comparable ORR activity to Pt/C and superior activity to the pristine Mn oxide electrodes. It was shown that the mass activity of synthesized hybrid materials not supported on carbon blacks outperforms the literature values of carbon-based materials. Mass activity performance was compared to the molar activity—calculated per mol of the Mn and Co oxides being in charge for the catalytic performance in oxygen reduction reaction. It was revealed that the synergic coupling of Mn oxides and $Co_3O_4$ have better catalytic performance in comparison to the electrodes based on pristine $MnO_2$ and $Co_3O_4/La_2O_3$. It was found that molar and mass activities give different information, since different amounts of active components are affecting the synergistic catalysis.

The crystal structure and morphological characteristics, as well as right amounts of investigated oxides, play the crucial role for high catalytic activity. Taking into account that the investigated materials are very low-cost materials especially compared to the state-of-the-art Pt/C-based electrodes, the demonstrated hybrid materials are promising catalysts for practical application for rechargeable metal–air batteries and fuel cells.

**Author Contributions:** M.V.: Investigation, methodology, writing—review and editing, writing—original draft. M.M.P.: Validation, visualization, writing—review and editing. S.E.P.: Conceptualization, formal analysis, data curation. M.M.: Conceptualization, formal analysis. M.R.P.P.: Formal analysis, visualization. S.S.: Conceptualization. B.F.: Funding acquisition. All authors have read and agreed to the published version of the manuscript.

**Funding:** This work was supported by the Ministry of Education, Science and Technological Development of the Republic of Serbia (Grant No. 451-03-9/2021-14/200026). The authors would like to thank the Ministry of Education, Science and Technological Development of the Republic of Serbia and DAAD, Germany, for funding of the Project No.: 57334757.

**Data Availability Statement:** All available data is contained within the article.

**Acknowledgments:** The authors would like to thank Tanja Barudžija and Miodrag Mitrić from Vinča Institute for support on the XRD measurements, Đorđe Veljović for SEM-EDS analysis. Special thanks to Vladimir Panić and Jasmina Stevanović for help in results analyses.

**Conflicts of Interest:** The authors declare that they have no known competing financial interest or personal relationships that could have appeared to influence the work reported in this paper.

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
