# Peer review of "Spray-Pyrolytic Tunable Structures of Mn Oxides-Based Composites for Electrocatalytic Activity Improvement in Oxygen Reduction"

_metals, doi:10.3390/met12010022_

Round 1

Reviewer 1 Report

This article reports the easy synthesis of Mn- Co- and La- composites and their performance as oxygen reduction catalysts. The results are interesting and the topic is of high relevance. The study methodology is sound and the conclusions are well supported by the results obtained. I recommend publication after some improvements:

1- The text could benefit from editing by a native speaker. In general it is easy to read and a really good effort by the authors, but in some instances it is not perfectly clear what the authors may be referring to, exactly. Ambiguity should be avoided in all cases and proofreading by a native speaker should sort the instances where the text can be clarified.

2-  A description/discussion of the rationale for the choice of materials and conditions synthesized and tested in this study is lacking. The introduction makes clear why Mn and Co were chosen, but there is no explanation as to why La was, and no discussion as to the structures and stoichiometries targetted. For example, on what basis were the relative amounts of Mn, Co and La chosen?

3- A more generalized discussion of Co-, Mn- and La-containing ORR catalysts for metal-air batteries (as the authors single this technology out) than what is provided should be included in the introduction. For instance, Co-catalysts have been explored for non-aqueous Li-Air systems, sometimes reporting 4e-/O2 stoichiometries (e.g. ACS Energy Lett. 2020, 5, 12, 3681–3691)

5- Figures 4 and 5 seem indistinguishable. Can the authors double check that the results are presented as described throughout the text?

Author Response

We would like to thank the reviewers for comments on our submitted manuscript. We have considered all points carefully and revised our manuscript accordingly. In the “revised manuscript” the modified text has been highlighted. Our answers in the response letter are marked in red.

Reviewer 1

Comments:
This article reports the easy synthesis of Mn- Co- and La- composites and their performance as oxygen reduction catalysts. The results are interesting and the topic is of high relevance. The study methodology is sound and the conclusions are well supported by the results obtained. I recommend publication after some improvements:

1-The text could benefit from editing by a native speaker. In general it is easy to read and a really good effort by the authors, but in some instances it is not perfectly clear what the authors may be referring to, exactly. Ambiguity should be avoided in all cases and proofreading by a native speaker should sort the instances where the text can be clarified.

Answer 1.1

Thank you for careful reading and helping us to improve the manuscript. The English has been corrected and improved. 

2-  A description/discussion of the rationale for the choice of materials and conditions synthesized and tested in this study is lacking. The introduction makes clear why Mn and Co were chosen, but there is no explanation as to why La was, and no discussion as to the structures and stoichiometries targetted. For example, on what basis were the relative amounts of Mn, Co and La chosen?

Answer 1.2

Indeed we have missed to explain the importance and utilization of La-based materials, as well as their mixed synergies. Thank you for pointing out this important issue. We have added new paragraph in the revised manuscript.

Shortly:

La-based perovskite material (LaBO3, B=Fe, Co, Ni, Mn, Cu etc.) as new class of materials in the mixed-oxide family have attracted increasing attention for potential replacement to the noble metals. They have shown promising catalytic performance for ORR in alkaline media. The activity of these La-based oxides strongly correlated with the covalent bond strength between B-site cation and the oxygenated species. La is located in the middle of the octahedral structure and plays stabilizing role. [1]

Additionally, considerable research has been devoted to evaluation of as electrocatalyst for ORR due to oxygen vacancies and interstitials with low oxygen vacancy energy, which makes the oxygen vacancies readily supplemented by rapid oxygen diffusion through the oxide and to the surface, leading to low activation energy. Furthermore, the existed interlayer defect structure of the oxides is also helpful to the active oxygen adsorption, which may have a positive effect on catalyzing the ORR. On the other side, even if lanthanum oxides have outstanding electronic structure, it is not electro-conductive, which limits its exertion in electro-catalysis and have the necessity of it combination with other oxides and carbon materials. [2,3]. Due to its high potential as stabilizing agent and high activity when mixed with other oxides it has been utilized in this work as well.

[1] https://doi.org/10.1021/acscatal.5b01667

[2] https://doi.org/10.1063/1.4832696

[3] https://doi.org/10.1021/jp506324j

The ratio of La:Co:Mn of 3:5:10 was chosen based on previous research [4,5], since we wanted to investigate the influence of Mn in mixture for ORR reaction. We have added the explanation of the chosen ratio in the revised manuscript as well.

[4] https://doi.org/10.1016/j.electacta.2019.134721.

[5]  https://doi.org/10.1016/j.jelechem.2021.115556.

3- A more generalized discussion of Co-, Mn- and La-containing ORR catalysts for metal-air batteries (as the authors single this technology out) than what is provided should be included in the introduction. For instance, Co-catalysts have been explored for non-aqueous Li-Air systems, sometimes reporting 4e-/O2 stoichiometries (e.g. ACS Energy Lett. 2020, 5, 12, 3681–3691)

Answer 1.3

Thank you for your suggestion. We have included the suggested manuscripts and as well as broaden the discussion related to usage of Co- Mn- La- for ORR in other systems. Please see the corrected manuscript.

4- Figures 4 and 5 seem indistinguishable. Can the authors double check that the results are presented as described throughout the text?

Answer 1.4

Thank you for careful observation. It is corrected as stated in manuscript: Figure 4 is related to the CV of the analyzed material in N2 atmosphere, while the Figure 5 presents the LSV results of the materials in the O2 atmosphere.

Reviewer 2 Report

In this manuscript, the synthesis of Co/Mn/La oxide hybrid materials was performed by single-step ultrasonic spray pyrolysis process and tested as electrocatalysts for oxygen reduction reaction (ORR). The comparison of molar catalytic activities points out the importance of the composition and that the synergy of Co and Mn is superior to Co3O4/La2O3 and pristine Mn oxide. However, the experiment in this manuscript is not enough to support the conclusions, and there is lack of in-depth analyzation for the results. Therefore, this manuscript is far away from being published. My specific comments are listed as follows:

  1. In conclusions, the authors stated that “It has been revealed that the synergic coupling of Mn oxides and Co3O4 have better catalytic performance in comparison to the electrodes based on pristine MnO2 and Co3O4/La2O3.” The synergistic coupling mechanism should be further discussed.
  2. Please indicate the role of lanthanum element in synthetic materials and its influence on the catalytic performance of the materials.
  3. Please calibrate the XRD peaks of the Co/La oxide hybrid material at positions 19° and 78°.
  4. On page 7, the authors stated that “the atomic ratio between lanthanum, cobalt and manganese is smaller than projected ratio. … as XRD analysis shows, La2O3 is presented in an amorphous state, which could also affect its apparently low content.” Please explain the reason in detail.
  5. On page 8, line 271, the authors stated that “Cobalt oxide particles dictate the CV behavior of Co3O4/La2O3 material in a way to resemble completely the redox processes of pure Co3O4.” Please add CV curve of pure Co3O4.
  6. It is suggested to supplement the element mapping of MnO2/Co3O4-500, MnO2/Co3O4-600 and Mn3O4/ Co3O4-800 to reveal the element distribution of materials.
  7. Figure 5 is the same as Figure 4. Please modify the picture to be consistent with the description of the manuscript.
  8. On page 10, line 313, the authors stated that “It follows that corresponding process(es) are not directly related to ORR, but to some reduction of the material induced by the presence of oxygen.” Please specify the content of “some”.
  9. In this manuscript, some references should be added to support the views. For example, the statement of “Although the synergic effect of MnO2 and Co3O4 oxides has been reported in many publications to increase ORR activity, the synergy of many parameters such as the right composition, crystalline structure and morphology has to be optimized in order to achieve it.” And it is suggested to give the sources of equation (1) and (2).
  10. In conclusions, the authors stated that “The electrochemical performance of MnOx/Co3O4 and Co3O4/La2O3 demonstrate a comparable ORR activity to Pt/C and superior activity to the pristine Mn oxide electrodes.” It is suggested to supplement the electrochemical test results of Pt/C electrode for comparison.
  11. There are some writing or expression errors in the manuscript. For example, on page 8, line 277, “MnO2/Co3O4-500, Fig. 2d” in parentheses should be corrected to “MnO2/Co3O4-600, Fig. 2d”, and the serial number of the XRD patterns should be changed to Figure 1. Please check them carefully and correct them.

Author Response

We would like to thank the reviewers for comments on our submitted manuscript. We have considered all points carefully and revised our manuscript accordingly. In the “revised manuscript” the modified text has been highlighted. Our answers in the response letter are marked in red.

Reviewer 2

Comments:
In this manuscript, the synthesis of Co/Mn/La oxide hybrid materials was performed by single-step ultrasonic spray pyrolysis process and tested as electrocatalysts for oxygen reduction reaction (ORR). The comparison of molar catalytic activities points out the importance of the composition and that the synergy of Co and Mn is superior to Co3O4/La2O3 and pristine Mn oxide. However, the experiment in this manuscript is not enough to support the conclusions, and there is lack of in-depth analyzation for the results. Therefore, this manuscript is far away from being published. My specific comments are listed as follows:

1. In conclusions, the authors stated that “It has been revealed that the synergic coupling of Mn oxides and Co3O4 have better catalytic performance in comparison to the electrodes based on pristine MnO2 and Co3O4/La2O3.” The synergistic coupling mechanism should be further discussed.

Answer 2.1

Although the synergy in ORR between Mn and Co oxides has been reported in literature, there are no clear proposition of some mechanism. The reviewer is right that following step in discussion should be the explanation of the most probable mechanism. However, as stated in revised manuscript, this requires additional investigation of the influence of parameters related to structure and composition on ORR, which is to be the scope of our further research activities.

2. Please indicate the role of lanthanum element in synthetic materials and its influence on the catalytic performance of the materials.

Answer 2.2

Indeed we have missed to explain the importance and utilization of La-based materials, as well as their mixed synergies. Thank you for pointing out this important issue. We have added new paragraph in the revised manuscript.

Shortly:

La-based perovskite material (LaBO3, B=Fe, Co, Ni, Mn, Cu etc.) as new class of materials in the mixed-oxide family have attracted increasing attention for potential replacement to the noble metals. They have shown promising catalytic performance for ORR in alkaline media. The activity of these La-based oxides strongly correlated with the covalent bond strength between B-site cation and the oxygenated species. La is located in the middle of the octahedral structure and plays stabilizing role. [1]

Additionally, considerable research has been devoted to evaluation of as electrocatalyst for ORR due to oxygen vacancies and interstitials with low oxygen vacancy energy, which makes the oxygen vacancies readily supplemented by rapid oxygen diffusion through the oxide and to the surface, leading to low activation energy. Furthermore, the existed interlayer defect structure of the oxides is also helpful to the active oxygen adsorption, which may have a positive effect on catalyzing the ORR. On the other side, even if lanthanum oxides have outstanding electronic structure, it is not electro-conductive, which limits its exertion in electro-catalysis and have the necessity of it combination with other oxides and carbon materials. [2,3]. Due to its high potential as stabilizing agent and high activity when mixed with other oxides it has been utilized in this work as well.

[1] https://doi.org/10.1021/acscatal.5b01667

[2] https://doi.org/10.1063/1.4832696

[3] https://doi.org/10.1021/jp506324j

3. Please calibrate the XRD peaks of the Co/La oxide hybrid material at positions 19° and 78°.

Answer 2.3

Thank you for your detailed observation. Indeed, we have missed these peaks. The peak on 19°correspond to the plane111 which correspond to the Co3O4 [1], while the peak at the position 78° correspond to the La2O3 [2]. The figure has been corrected accordingly, please see the revised manuscript.

[1] https://iopscience.iop.org/article/10.1088/1757-899X/202/1/012066

4. On page 7, the authors stated that “the atomic ratio between lanthanum, cobalt and manganese is smaller than projected ratio. … as XRD analysis shows, La2O3 is presented in an amorphous state, which could also affect its apparently low content.” Please explain the reason in detail.

Answer 2.4

Additional considerations are incorporated in the revised manuscript.

5. On page 8, line 271, the authors stated that “Cobalt oxide particles dictate the CV behavior of Co3O4/La2O3 material in a way to resemble completely the redox processes of pure Co3O4.” Please add CV curve of pure Co3O4.

Answer 2.5

The cyclic voltammetry performance of Co3O4 has been already reported in many literatures. In our manuscript, we have referred to the literature results [3] that are reporting behavior of Co3O4 on GC which is highly comparable to our conditions. Since, the behavior is quite similar we have decided to cite the following manuscript. Please see the revised manuscript.

[3] http://doi:10.1007/s10008-017-3862-2

6. It is suggested to supplement the element mapping of MnO2/Co3O4-500, MnO2/Co3O4-600 and Mn3O4/ Co3O4-800 to reveal the element distribution of materials.

Answer 2.6

Additional explanations are incorporated in the revised manuscript.

7. Figure 5 is the same as Figure 4. Please modify the picture to be consistent with the description of the manuscript.

Answer 2.7

Thank you for careful observation. Please see the revised manuscript.

8. On page 10, line 313, the authors stated that “It follows that corresponding process(es) are not directly related to ORR, but to some reduction of the material induced by the presence of oxygen.” Please specify the content of “some”.

Answer 2.8

This is referring to the reported ORR mechanism on Mn-oxide that goes through additional step of MnO2 to MnOOH, presented by equations 3-5b in the manuscript. However, we agree that the formulation of “some” has to be corrected, and it was exchange in the revised manuscript.

9. In this manuscript, some references should be added to support the views. For example, the statement of “Although the synergic effect of MnO2 and Co3O4 oxides has been reported in many publications to increase ORR activity, the synergy of many parameters such as the right composition, crystalline structure and morphology has to be optimized in order to achieve it.” And it is suggested to give the sources of equation (1) and (2).

Answer 2.9

The references have been added.

10. In conclusions, the authors stated that “The electrochemical performance of MnOx/Co3O4 and Co3O4/La2O3 demonstrate a comparable ORR activity to Pt/C and superior activity to the pristine Mn oxide electrodes.” It is suggested to supplement the electrochemical test results of Pt/C electrode for comparison.

Answer 2.10

Since the Pt is not in the interest of our research due to high price and scarcity as main limitations for the application in the electrochemical devices, and taking into account that there are broad literature and many manuscripts related to the analysis of the Pt-materials, we have referred to the cited papers for the comparisons instead of testing the Pt/C material.

11. There are some writing or expression errors in the manuscript. For example, on page 8, line 277, “MnO2/Co3O4-500, Fig. 2d” in parentheses should be corrected to “MnO2/Co3O4-600, Fig. 2d”, and the serial number of the XRD patterns should be changed to Figure 1. Please check them carefully and correct them.

Answer 2.11

Thank you for careful reading and helping us to improve the manuscript. The errors have been corrected in the revised manuscript.

Reviewer 3 Report

Thanks for the invitation to review the manuscript. Authors prepared various Mn based compounds and investigated their ORR activity in alkaline solution. The manuscript has not been organized well and many things require proper explanation.

  1. In introduction it is not clear why Mn based materials should be combined with other metal oxide such as Co and La which can eventually increase the cost of electrocatalyst.
  2. Figure 1 is assigned as Figure 2.
  3. Figure 4 and 5 appears to be same. Also, most of electrocatalytic activity has been reported on RHE so authors should also convert SCE to RHE for better understanding and comparision with other catalyst.
  4. The ORR performance curves is not normal and there is no limiting current in most of catalyst. Why?
  5. Outdated KL equation should not been used to determine electron transfer number.
  6. Related referecnes should be included Hui Liu et al 2021 J. Electrochem. Soc. 168 080518; NPG Asia Mater., 2016, 8, e294;   https://doi.org/10.1002/smll.201804958

Author Response

We would like to thank the reviewers for comments on our submitted manuscript. We have considered all points carefully and revised our manuscript accordingly. In the “revised manuscript” the modified text has been highlighted. Our answers in the response letter are marked in red.

Reviewer 3

Comments:
Thanks for the invitation to review the manuscript. Authors prepared various Mn based compounds and investigated their ORR activity in alkaline solution. The manuscript has not been organized well and many things require proper explanation.

1. In introduction it is not clear why Mn based materials should be combined with other metal oxide such as Co and La which can eventually increase the cost of electrocatalyst.

Answer 3.1

The explanation for the synergy of the oxide materials has been additionally added. Please see introduction in the revised manuscript.

 2. Figure 1 is assigned as Figure 2.

Answer 3.2

Thank you. It has been corrected.

3. Figure 4 and 5 appears to be same. Also, most of electrocatalytic activity has been reported on RHE so authors should also convert SCE to RHE for better understanding and comparision with other catalyst.

Answer 3.3

Thank you for careful observation. It is corrected as stated in manuscript: Figure 4 is related to the CV of the analyzed material in nitrogen atmosphere, while the Figure 5 presents the LSV results of the materials in the oxygen atmosphere and it is corrected.

The SCE has been corrected to the RHE.

4. The ORR performance curves is not normal and there is no limiting current in most of catalyst. Why?

Answer 3.4

The limiting current is normal and it is appearing only for the electrochemical reactions that are limited by the reactant concentration (mass-controlled reactions) and here that is not the case.

5. Outdated KL equation should not been used to determine electron transfer number.

Answer 3.5

In general, Koutecky–Levich equation is well known equation which analysis allows obtaining critical kinetic parameters for the redox reactions.

Please find below the latest manuscripts published in the last years that are using the KL equations as well.

For example:

The manuscript publish in 2018

https://doi.org/10.1007/s10008-017-3862-2

The manuscript publish in 2020

https://doi.org/10.1016/j.electacta.2020.136592

The n value as well calculated in this manuscript from 2021 is using following eq. as well.

https://doi.org/10.1016/j.jallcom.2020.157012

6. Related referecnes should be included Hui Liu et al 2021 J. Electrochem. Soc. 168 080518; NPG Asia Mater., 2016, 8, e294;   https://doi.org/10.1002/smll.201804958

Answer 3.6

Thank you for your suggestion, it has been included in the manuscript . Please check the corrected version of the manuscript.

Round 2

Reviewer 2 Report

This manuscript is advised to be accepted since all the problems have been addressed.

Reviewer 3 Report

Authors have revised manuscript well therefore it can be accepted in its current form.